# Efficient Self-Improvement in Multimodal Large Language Models: A Model-Level Judge-Free Approach

**Shijian Deng**[1] **Wentian Zhao**[2] **Yu-Jhe Li**[2] **Kun Wan**[2]
**Daniel Miranda**[2] **Ajinkya Kale**[2] **Yapeng Tian**[1]
[1]The University of Texas at Dallas [2]Adobe Inc.
{shijian.deng, yapeng.tian}@utdallas.edu,
{wezhao, jhel, kuwan, miranda, akale}@adobe.com

## Abstract

Self-improvement in multimodal large language models (MLLMs) is crucial for enhancing their reliability and robustness. However, current methods often rely heavily on MLLMs themselves as judges, leading to high computational costs and potential pitfalls like reward hacking and model collapse. This paper introduces a novel, model-level judge-free self-improvement framework. Our approach employs a controlled feedback mechanism while eliminating the need for MLLMs in the verification loop. We generate preference learning pairs using a controllable hallucination mechanism and optimize data quality by leveraging lightweight, contrastive language-image encoders to evaluate and reverse pairs when necessary. Evaluations across public benchmarks and our newly introduced IC dataset, designed to challenge hallucination control, demonstrate that our model outperforms conventional techniques. We achieve superior precision and recall with significantly lower computational demands. This method offers an efficient pathway to scalable self-improvement in MLLMs, balancing performance gains with reduced resource requirements. The code is available at https://github.com/ShijianDeng/ESI-MLLM.

## 1 Introduction

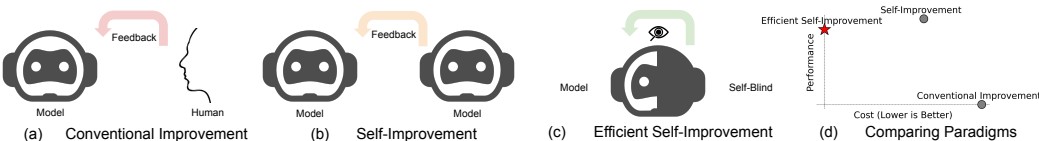

Figure 1: Comparison of three different improvement paradigms. (a) The conventional improvement paradigm requires humans to annotate feedback data and feed it into the model for improvement, making it the least efficient approach. (b) The self-improvement paradigm leverages the model itself to provide feedback; however, this approach is still inefficient due to the high cost and potential bias of using large models as verifiers. (c) Our efficient self-improvement paradigm improves the model without human feedback or model-level self-feedback by using a predefined data generation strategy combined with a lightweight verifier, achieving both efficiency and performance improvement. (d) Among all three paradigms, our approach offers the best trade-off between performance and cost.

Self-improvement is a natural way for humans to learn independently, enabling them to acquire knowledge and skills beyond what they learn from their teachers. This same paradigm is being gradually adapted for large language models (LLMs) and multimodal large language models (MLLMs) to achieve performance improvements beyond the seed model with minimal human supervision.

Recent studies have explored various approaches (Favero et al., 2024; Zhou et al., 2024; Deng et al., 2024; Yu et al., 2024b) for self-improvement in MLLMs. For instance, RLAIF-V (Yu et al., 2024b) uses MLLMs to evaluate and score responses generated by another MLLM, creating preference learning pairs from responses to the same image and question. M3ID (Favero et al., 2024), POVID (Zhou et al., 2024), and STIC (Deng et al., 2024) employ techniques like bad prompts, image corruption, unconditioned generation, and response injection to generate hallucinated responses as negative samples for preference learning.

However, several issues limit this paradigm: 1) it relies heavily on the quality of the verifier (e.g., a reward model); 2) the process can be resource-intensive, generating numerous samples but only using a tiny subset; 3) the cost multiplies when another large model is needed for verification, especially when generating reasoning or comments for final evaluation. Past studies (Valmeekam et al., 2024a;b) have underscored the necessity of an external verifier.

To overcome these challenges, we propose an alternative approach, illustrated in Fig 1, enabling self-improvement without directly using an MLLM as a verifier for dataset filtering. Our method involves controlled hallucination to generate preference learning pairs, lightweight evaluation with a contrastive language-image encoder to optimize data quality, and direct preference optimization (DPO) (Rafailov et al., 2024) to train the seed model.

First, we employ an efficient, controllable approach to generate simple negative or hard-negative samples, thereby creating the initial preference learning pairs. We use a controller that ranges from 0 to 1 to regulate the level of hallucination in responses. After generating the initial dataset, we leverage a lightweight, contrastive language-image pretrained encoder to compute the average sentence-level CLIPScore (Hessel et al., 2021). This score is used to identify and update pairs in which the negative sample's score exceeds that of the positive sample by a set threshold, thereby refining our preference learning dataset. Finally, we use the optimized dataset to train the seed model via DPO, resulting in a self-improved model. Extensive evaluations on both in-house and public benchmarks demonstrate significant gains over the original seed model.

Our primary contributions are as follows: 1) We propose a novel and efficient framework for self-improvement in MLLMs that: (a) combines a predefined, controllable mechanism for efficient negative sample generation, and (b) uses a lightweight verifier to effectively control positive and negative pairs, automatically reversing them when necessary. 2) We collected a new IC dataset, which includes GPT-4o assisted evaluation of both precision and recall of MLLMs. 3) Experimental results demonstrate that we achieve significantly better performance over the seed model on both our IC dataset and other public datasets.

## 2 Related Work

### 2.1 Multimodal Large Language Models

To leverage the knowledge and reasoning capabilities of LLMs in multimodal settings and address broad multimodal comprehension challenges, MLLMs have been developed. Significant work has been done in this field, such as LLaVA (Liu et al., 2024c), which connects CLIP with the LLaMA model through an adapter; Qwen-VL (Bai et al., 2023), which implements grounding and text-reading abilities by aligning image-caption-box tuples; CogVLM (Wang et al., 2023b), which uses a trainable visual expert module in the attention and FFN layers to enable deep fusion of vision-language features without sacrificing NLP task performance; InternVL (Chen et al., 2024b), which employs both contrastive and generative tasks to better align the large-scale vision foundation model with MLLM; Pixtral (Agrawal et al., 2024), which processes images through the vision encoder at their native resolution and aspect ratio, converting them into image tokens for each patch in the image, allowing it to handle any number of images of arbitrary sizes in its large context window; and LLaMA3.2 Vision, which incorporates visual-recognition capabilities into LLaMA 3 (Dubey et al., 2024) via a compositional approach to ensure that text-only task performance is not affected by the addition of visual-recognition capabilities.

## 2.2 Self-Improvement

Even after large-scale pretraining, instruction tuning, and reinforcement learning from human feedback (RLHF) (Ouyang et al., 2022), large models may still show vulnerabilities in various cases. Although new data can always be prepared to improve a specific missing capability of the model, this is not a sustainable long-term solution to fix all issues. To enhance a large model's helpfulness and trustworthiness without exhausting human effort, a new self-improvement paradigm has been adopted, as systematically discussed in the survey (Tao et al., 2024). For MLLMs, this often involves two key steps: sampling and verification.

**Sampling.** To improve the seed model's performance, the first step is to sample the necessary data. The simplest approach is to change seeds and randomly sample a large number of outputs, though this may not be efficient. Instead, users can predefine the type of data to generate by employing improved prompts and chains of thought to produce high-quality data, or by using corrupted images, attention masks, and text to generate negative data, as explored in POVID (Zhou et al., 2024), STIC (Deng et al., 2024), and BDHS (Amirloo et al., 2024). In M3ID (Favero et al., 2024), the authors also use mutual information from information theory to better control the quality of generated outputs and therefore achieve more effective sampling. In our work, we further simplify this sampling approach, making the process more practical for optimizing the margin (Deng et al., 2025) in preference learning.

**Verification.** The model would not significantly improve if it simply reuses any generated data for retraining. A more effective approach is to perform data selection before training. There are many ways to achieve this. The simplest method is majority voting, though this may fail when the correct output is not the most common. A verifier, while optional, is commonly used as an additional quality control layer for data. The most straightforward and widely used verification method is to use an MLLM as a reward model, as seen in RLAIF-V (Yu et al., 2024b). However, this approach has limitations related to cost and potential bias due to the reward models' own limitations. An external verifier can help address these issues. For example, CLIP-DPO (Ouali et al., 2024) utilizes CLIP to rank short descriptions generated by the MLLM. We adopted a similar approach and extended it to suit long captions, seamlessly integrating it into our self-improvement framework along with our sampling methods to further enhance the robustness of our pipeline.

## 3 Method

This section describes our approach to efficient self-improvement in MLLMs. We begin with a description of our controllable method for generating positive and negative data pairs for training. Next, we highlight the importance of incorporating a lightweight quality control mechanism to ensure that the generated data effectively guides the learning process. Finally, we explain how the generated data is used to train the seed model with DPO, culminating in a self-improved model.

### 3.1 Motivation

To perform well, preference learning requires diverse data and accurate preference labels for each pair, making it critical to establish a fully controllable approach for generating the required dataset. While it is challenging to produce data that surpasses the quality of what the seed model can generate, it is relatively feasible to create data that is worse than what the model can typically produce. It is also important to know how much worse the sample we need before we generate it since both too hard or too simple pairs may not work best. Based on these observations, we propose a simple yet efficient method for generating preference learning data pairs with any difference level between positive samples and negative samples.

The high computational cost of running models with a large number of parameters, combined with the inherent inductive biases of MLLMs, imposes significant limitations on

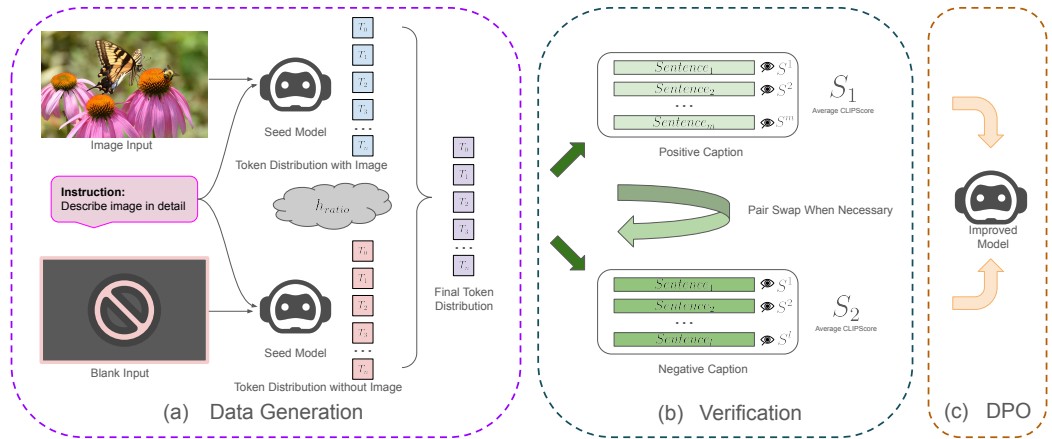

Figure 2: Overview of our framework. Our efficient self-improvement framework combines two main strategies: (a) We use a simple yet effective predefined preference dataset generation approach, employing two decoding paths during response generation. By adjusting the hallucination ratio $h_{\text{ratio}}$, we can control whether a negative or positive sample is generated for preference learning. (b) After the initial preferences are generated, we use a lightweight contrastive language-image pretrained encoder to calculate the average sentence-level CLIP_score difference between the initial positive and negative samples, swapping them when necessary to ensure the quality of the final preference dataset. (c) Finally, we apply DPO with the resulting dataset to improve the model.

relying on large models as verifiers. To address these challenges, we introduce an objective and lightweight alternative for verification purposes.

## 3.2 Controllable Dataset Generation

To train a self-improving model with preference learning, we first need to prepare a suitable dataset. To generate preference pairs, we use the seed model $m_0$ to create a positive response $y_w$ and a negative response $y_l$ from the same input image $x_{img}$ and instruction $x_{instruct}$. To obtain the negative sample, we introduce interventions during the decoding process of the MLLM. We use two decoding paths: a conditional path $p_c$ that generates a response based on both the input image $x_{img}$ and instruction $x_{instruct}$, and an unconditional path $p_u$ that uses only the instruction $x_{instruct}$, without the image $x_{img}$. The generation is controlled by the hallucination ratio, i.e., $h_{\text{ratio}}$, which determines the level of hallucination to be injected into the generated caption, ranging from 0 to 1. A higher $h_{\text{ratio}}$ denotes injecting more hallucinations into the response.

As shown in Fig. 2, for each output token, the distribution is determined by combining the token distribution $t_c$ from the conditional path and the token distribution $t_u$ from the unconditional path, weighted by the hallucination ratio (i.e., $h_{\text{ratio}}$):

$$t = (1 - h_{\text{ratio}}) \cdot t_c + h_{\text{ratio}} \cdot t_u. \tag{1}$$

The two paths, $p_c$ and $p_u$, do not interact, ensuring that the unconditional path never accesses any information from the input image, thus serving as a "pure" hallucination source. Each pair is initially labeled, with the response generated under the lower $h_{\text{ratio}}$ assigned as positive and the other as negative. The $h_{\text{ratio}}$ follows a predefined distribution, such as uniform or Gaussian, and remains fixed for each decoding process once assigned.

## 3.3 Lightweight Preference Data Inversion

Although the generated pairs initially have assigned positive or negative labels, these labels may not always be accurate, as the conditional generation process with the seed MLLM can sometimes introduce a certain level of hallucination in the decoded text. To address this, we

---

**Algorithm 1** Efficient Self-Improving MLLM with Preference Learning

---

**Require:** Seed model $m_0$, dataset $\{(x^i_{\text{img}}, x^i_{\text{instruct}})\}^N_{i=1}$
**Ensure:** Improved model $m_1$
1: **for** each $(x_{\text{img}}, x_{\text{instruct}})$ in dataset **do**
2:     Sample hallucination ratio $h_{\text{ratio}} \in [0,1]$ from a predefined distribution
3:     **for** each time step $t$ **do**
4:         Compute conditional token distribution $t_c = p_c(y_t \mid x_{\text{img}}, x_{\text{instruct}}, y_{<t})$
5:         Compute unconditional token distribution $t_u = p_u(y_t \mid x_{\text{instruct}}, y_{<t})$
6:         Compute final token distribution $t = (1 - h_{\text{ratio}}) \times t_c + h_{\text{ratio}} \times t_u$
7:         Sample token $y_t \sim t$
8:     **end for**
9:     Obtain responses $y_{\text{low}}$ (lower $h_{\text{ratio}}$) and $y_{\text{high}}$ (higher $h_{\text{ratio}}$)
10:    Assign initial labels: positive response $y^i_w = y_{\text{low}}$, negative response $y^i_l = y_{\text{high}}$
11:    Compute average CLIP scores $CLIP\_score^i_w$ and $CLIP\_score^i_l$
12:    **if** $CLIP\_score^i_w - CLIP\_score^i_l < threshold$ **then**
13:       Swap $y^i_w$ and $y^i_l$
14:    **end if**
15:    Add final pair $(x_{\text{img}}, x_{\text{instruct}}, y^f_w, y^f_l)$ to preference dataset $D$
16: **end for**
17: Select subset $D_{\text{sub}}$ from $D$ based on $CLIP\_score$ difference
18: Initialize improved model $m_1 \leftarrow m_0$
19: **for** each $(x, y^w, y^l)$ in $D_{\text{sub}}$ **do**
20:    Compute $\Delta(x, y^w, y^l; \theta) = [\log \pi_\theta(y^w \mid x) - \log \pi_\theta(y^l \mid x)] - [\log \pi_0(y^w \mid x) - \log \pi_0(y^l \mid x)]$
21:    Compute loss $L(\theta) = -\log \sigma \left( \Delta(x, y^w, y^l; \theta) \right)$
22:    Update model parameters $\theta$ by minimizing $L(\theta)$
23: **end for**

---

implement an additional quality control step to manage cases where initial labeling may be incorrect.

Specifically, we use a lightweight CLIP model, which is the vision-language contrastive pretrained encoder of the MLLM. For each initial pair $(y^i_w, y^i_l)$, we calculate the CLIP_score between the image and each decoded sentence. Since CLIP has a 77-token limit and cannot accommodate overly long captions, we compute the average sentence-level CLIP_scores for the initial positive caption, $CLIP\_score^i_w$, and the initial negative caption, $CLIP\_score^i_l$. If $CLIP\_score^i_w - CLIP\_score^i_l < threshold$, indicating that the initial positive is rated much lower than the initial negative, we swap the preference labels, designating $y^i_w$ as the final negative $y^f_l$ and $y^i_l$ as the final positive $y^f_w$. Otherwise, we retain the original order in the final pair.

This process prevents cases where an initial negative sample might outperform its counterpart, which could undermine subsequent preference learning. After this step, we obtain the final preference pairs $(y^f_w, y^f_l)$, which are used in preference alignment training to improve the seed model $m_0$.

### 3.4 Preference Learning Finetuning

After obtaining the final pairs of positive caption $y^f_w$ and negative caption $y^f_l$ generated from the same input image $x_{img}$ and instruction $x_{instruct}$, we select a subset of the preference dataset $D$ within a certain range of informativeness (Wu et al., 2024), defined by the CLIP_score difference: $CLIP\_score^f_w - CLIP\_score^f_l$, forming $D_{sub}$. We then use DPO, a commonly used, low-cost alternative to PPO, to train the seed model $m_0$, further enhancing its performance. Through this finetuning process, we obtain an improved model $m_1$, which is self-improved from the seed model $m_0$ using its own generated dataset. The detailed process is illustrated in Fig. 2 and Algorithm 1.

## 4 Experiments

To evaluate the effectiveness of our proposed self-improvement framework, we tested it on both our IC dataset using GPT-4o series evaluation and other commonly used benchmarks. We introduce the experimental settings for dataset generation and verification, followed by

a detailed analysis of results and ablation studies to demonstrate the effectiveness of our framework and each of its design modules.

## 4.1 Datasets

**IC Dataset.** Current hallucination benchmarks primarily evaluate the precision of captions while often ignoring recall. To comprehensively assess MLLMs' captioning abilities, we have collected a new dataset containing 150 challenging images prone to hallucination across a wide range of domains and scenarios. These include abstract concepts, animals, animations, artistic content, common sense violations, documents, events, fashion, food, handwriting, illustrations, objects, people, posters, scenes, technology, and vehicles.

After generating captions, we use the GPT-4o series to evaluate them based on precision (elements in the caption that are present in the image) and recall (elements in the image that are captured in the caption) to calculate a final F1 score, which serves as a measure of caption quality. Some examples are shown in Fig. 6.

**Public Benchmarks.** Besides our newly collected IC dataset, we also selected Object Hal-Bench (Rohrbach et al., 2018), a classic benchmark that focuses on evaluating object-level hallucination in vision-language models. To further explore the generalizability of our framework, we incorporated Qwen2-VL (Wang et al., 2024), along with LLaVA, to evaluate their performance on additional benchmarks, including AMBER (Wang et al., 2023a), MMHal-Bench (Sun et al., 2023), MMStar (Chen et al., 2024a), GQA (Hudson & Manning, 2019), OCRVQA (Mishra et al., 2019), MathVista (Lu et al., 2024), and ChartQA (Masry et al., 2022). These benchmarks cover various tasks such as VQA, OCR, math, and chart understanding.

## 4.2 Experiment Setup

For the seed model $m_0$, we used LLaVA-1.5-13B (Liu et al., 2024a), a popular and representative MLLM. An 8xA100 node with 80GB VRAM per GPU was used for DPO training, while data generation and other processes were performed on a single GPU. We trained the model for $2,672$ steps with a learning rate of $5e - 7$.

During data generation, we sampled 100k images from the LLaVA instruction tuning dataset, llava_v1_5_mix665k, and removed all question-answer pairs. Using the prompt "Describe image in detail," the model generated responses with an $h_{ratio}$ ranging from 0 to 1. Initially, captions generated with a lower $h_{ratio}$ were assigned as the initial positive samples, $y_w^i$, while captions generated from the same inputs $x_{instruct}$ and $x_{img}$ with a higher $h_{ratio}$ were assigned as the initial negative samples, $y_l^i$. This process resulted in 100k initial preference pairs.

For the obtained caption pairs, each sentence was extracted, and the CLIP model was used to compute the CLIP_score for each image-caption pair. For sentences longer than the CLIP model's context limit, we split them into shorter sub-sentences, computed their respective CLIP_scores, and calculated the average CLIP_score for each caption by averaging the scores from all sentences and sub-sentences. If the average CLIP_score of a negative caption was higher than that of the positive caption, we swapped the positive and negative samples. The pairs were then sorted by CLIP_score difference, from low to high, and organized into 10 splits, each containing 10k pairs. For each split, we trained a LLaVA model and conducted inference on the IC dataset and other benchmarks to gather results. For the IC dataset, GPT-4o was used as the evaluator to compute precision, recall, and F1 score.

## 4.3 Results

With the improved model $m_1$ derived from the original seed model $m_0$, we evaluated performance across different benchmarks and presented the results in Tab 1 and Tab 2. As shown, the self-improved model outperforms previous models on both benchmarks. In particular, compared to the original seed model LLaVA-1.5-13B, performance has improved significantly, clearly demonstrating the effectiveness of our framework. Compared to

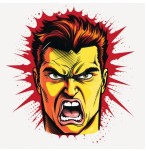 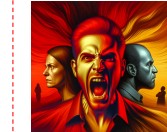 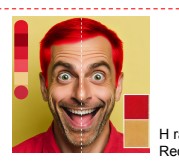 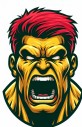

Original | LLaVA 1.5 13b Reconstruct | H ratio: 0.1 Reconstruct | **Ours Reconstruct**

Figure 3: Image reconstruction examples. To further demonstrate the effectiveness of our training framework, we use a text-to-image diffusion model, DALL·E 3, to convert captions generated by the models back into images. The reconstructed image from the original model's caption contains significant hallucination, while increasing the $h_{\text{ratio}}$ during generation produces a negative caption that, when reconstructed, shows even more hallucination in attributes like style and emotion. However, after training with these generated caption pairs, the reconstructed image from the improved model's caption closely resembles the original, surpassing both the positive and negative samples.

| Model | Size | Feedback | Object HallBench | |
|---|---|---|---|---|
| | | | Resp. ↓ | Ment. ↓ |
| VCD (Leng et al., 2024) | 7B | No | 48.8 | 24.3 |
| OPERA (Huang et al., 2024) | 7B | No | 45.1 | 22.3 |
| Less-is-more (Yue et al., 2024) | 7B | No | 40.3 | 17.8 |
| LURE (Zhou et al., 2023) | 7B | No | 27.7 | 17.3 |
| QWEN-VL (Bai et al., 2023) | 10B | No | 40.4 | 20.7 |
| MiniGemini (Li et al., 2024) | 34B | No | 14.5 | 8.0 |
| LLaVA-NeXT (Liu et al., 2024b) | 34B | No | 12.6 | 6.4 |
| HA-DPO (Zhao et al., 2023) | 7B | Rule | 39.9 | 19.9 |
| POVID (Zhou et al., 2024) | 7B | Rule | 48.1 | 24.4 |
| Silkie (Li et al., 2023) | 10B | GPT-4V | 27.1 | 13.4 |
| LLaVA-RLHF (Sun et al., 2023) | 13B | Human | 38.1 | 18.9 |
| RLHF-V (Yu et al., 2024a) | 13B | Human | 12.2 | 7.5 |
| LLaVA-1.5 (Liu et al., 2024a) | 7B | No | 53.6 | 25.2 |
| LLaVA-1.5 (Liu et al., 2024a) | 13B | No | 51.6 | 24.6 |
| + Ours | 13B | Self-Efficiency | **9.4** | **5.1** |

Table 1: Main results of our experiments on Object HallBench. Comparison of various models across different metrics. Resp. indicates the response-level metric, and Ment. represents the mention-level metric. The best results are **highlighted**.

previous methods, ours is the first to emphasize an efficient self-improvement approach that balances efficiency and effectiveness.

With different CLIP_score difference pairs generated with $h_{\text{ratio}}$ sampled from a Gaussian distribution, we also present the ablation study results in Fig. 4. We observe a clear performance gain for each component added to our framework, compared to the seed model $m_0$ and the model without that component, as shown in Fig. 5.

In Fig. 6, we show qualitative results to demonstrate the differences between our self-improved model and the original seed model. We also use the generated captions to perform image reconstruction with the DALL·E 3 model, as shown in Fig. 3. For other benchmarks, results are presented in Tab. 3.

### 4.4 Experimental Analysis

From the comprehensive evaluation results, we observe that our self-improved model shows significant performance gains across various benchmarks compared to the initial seed model in all evaluation dimensions. Here are some detailed discussions and findings from our experiments:

**Our model performs substantially better than the initial seed model,** as demonstrated by both quantitative and qualitative results. In Tab. 1, our model achieves scores of 9.4 for object response level and 5.1 for mention level, ranking it among the best of all models. These results on a popular public benchmark for evaluating hallucination demonstrate that our model outperforms all others at multiple evaluation levels, despite not using additional

| Model | Precision ↑ | Recall ↑ | F1 ↑ | Change ↑ |
|---|---|---|---|---|
| LLaVA-1.5-13B | 6.6 | 6.56 | 6.58 | 0.00 |
| + random negative samples DPO | 6.58 | 6.79 | 6.68 | 0.10 |
| + Ours | **7.74** | **7.78** | **7.76** | **1.18** |

Table 2: Main results on the IC dataset. For the random negative samples DPO method, the model generates random captions without an image to serve as negative samples; these are paired with the original captions (positive samples) to train an improved model using DPO. **Precision** measures how many elements in the caption are also in the image (higher scores indicate lower hallucination in the caption). **Recall** measures how many elements in the image are included in the caption, providing a complementary metric for hallucination evaluation. **F1** is the harmonic mean of precision and recall. All scores are on a scale from 1 (worst) to 10 (best). The best scores are highlighted.

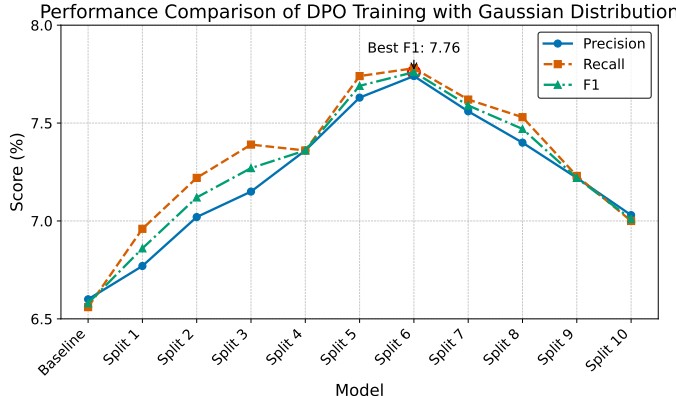

Figure 4: Performance comparison of DPO training using various CLIP_score differences generated with $h_{\text{ratio}}$ sampled from a Gaussian distribution and ranked from low to high. For these experiments, we set $\mu = 0.5$ and $\sigma = 0.15$.

human feedback during finetuning or requiring feedback from an MLLM. Instead, it relies on a lightweight CLIP encoder, highlighting the efficiency and effectiveness of our proposed framework.

**Each of our designed modules contributes to performance improvement.** As shown in Fig. 5, using $h_{\text{ratio}}$ to generate preference learning pairs and fine-tuning the seed model with DPO result in a substantial gain in both precision and recall compared to the original model. Adding CLIP_score difference filtering to exclude pairs with negative differences further enhances the model's performance. Instead of discarding these pairs, swapping the positive and negative samples when their CLIP_score difference is negative leads to another notable improvement. This highlights the necessity of a lightweight, post-hoc guardrail with CLIP_score difference to further boost performance. Finally, rather than using a fixed $h_{\text{ratio}}$, we experimented with a more diverse approach by randomly sampling $h_{\text{ratio}}$ values from a Gaussian distribution. This method introduces greater diversity, likely because it encompasses a broader range of cases, making the dataset more generalizable. This approach further improves the model's performance by increasing the variety of negative samples.

**The visual-language correspondence difference matters.** To evaluate model performance across settings with different preference pair combinations, we conducted extensive experiments varying the average sentence-level CLIP_score differences using Gaussian distribution sampling (see Fig. 4, highlighting structured variability). The results indicate that selecting an optimal CLIP_score difference is critical, with performance peaking when differences are moderate: neither too large nor too small. This aligns with human learning patterns, where understanding improves most when examples are distinct enough to differentiate yet similar enough to allow meaningful comparisons. This finding also matches the data selection theory in recent research (Deng et al., 2025).

|  | AMBER | HallusionBench | MMStar | GQA | OCRVQA | MathVista | ChartQA |
|---|---|---|---|---|---|---|---|
| LLaVA-1.5-13B | 82.09 | 50.47 | 0.36 | 53.58 | 59.96 | 26.70 | 13.48 |
| + Ours | **82.35** | **50.79** | 0.36 | **54.08** | **60.25** | **28.00** | **13.84** |
| Qwen2VL-2B | 82.79 | 61.83 | **0.48** | 60.44 | 68.75 | 47.60 | 73.36 |
| + Ours | **83.41** | **62.46** | 0.47 | 59.87 | **68.98** | **48.00** | **73.88** |
| Qwen2VL-7B | 85.67 | 67.61 | 0.60 | 62.14 | 71.61 | 59.80 | 81.12 |
| + Ours | **85.81** | **68.03** | 0.60 | **62.47** | **71.91** | **60.10** | **81.24** |

Table 3: Performance comparison with more models across other benchmarks. Best performance for each model is **highlighted**.

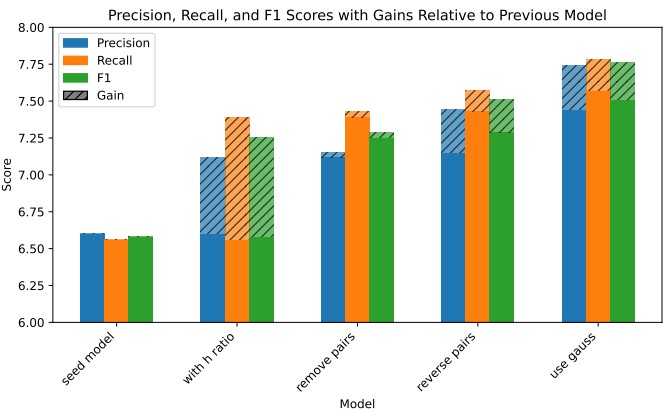

Figure 5: Comparison of model performance when adding certain mechanisms in our framework. The striped sections indicate the performance gain after adding each module. We use LLaVA-1.5-13B as the seed model. The modules gradually added to the seed model are: using $h_{\text{ratio}}$, removing pairs with negative CLIP_score differences, swapping pairs instead of removing them, and adding Gaussian distribution sampling. Each module in our design contributes to the final performance.

**Hallucination reduction and better reconstructions.** From the qualitative examples in Fig. 6 and the reconstruction results in Fig. 3, we observe that while the seed model tends to hallucinate significantly, our model generates far more accurate content when provided with the same image and text prompt. These results also demonstrate how hallucinations can impair the reconstruction of an original image given a caption from a hallucination-prone model, and how our approach mitigates this issue. This could potentially contribute to building better reconstruction or generation models by using captions generated by our model.

**Generalization to Other Tasks.** As we observed in Tab. 3, the results confirm that our self-improved model generalizes well across a wide range of tasks, including OCR, VQA, ChartQA, and even for visual reasoning tasks such as MathVista. This demonstrates that the gains from our framework can benefit a broader domain.

## 5 Limitations and Future Work

Although our experiments demonstrate that our framework is highly effective for enhancing the initial model's performance, we acknowledge some limitations and highlight areas for exploration and improvement in future work.

**Advanced Guardrail.** Our framework's primary reward signal operates on the assumption that a higher $h_{\text{ratio}}$ indicates a worse response. CLIP_score serves a secondary role as a guardrail to handle instances where this assumption fails. Consequently, a more robust guardrail than CLIP_score would ultimately improve the model's performance.

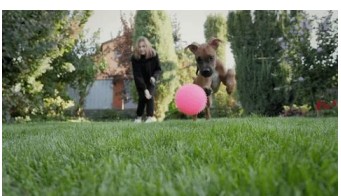
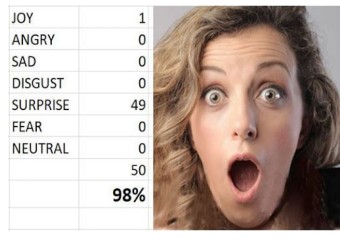

Figure 6: Examples of qualitative results. With the same input image and instruction prompt "Describe image in detail", the caption generated by the original seed model, LLaVA-1.5-13B, contains many hallucinations. In contrast, the model trained through our efficient self-improvement framework describes the image accurately, without hallucinated content. Hallucinated content is highlighted in red, and accurate content is highlighted in blue for easy identification.

**Recursive Self-Improvement.** Due to limited resources, we were unable to investigate whether recursive self-improvement is feasible by iteratively applying our framework in multiple rounds, from data generation to preference learning finetuning, to go beyond $m_1$ and potentially achieve $m_2$, $m_3$, and so on. This could reveal whether further improvements are possible or if an upper performance bound exists.

**Scaling with Larger Models and Datasets.** Because of training costs, we were unable to experiment with even larger models or larger datasets. Exploring the scaling laws of the framework with additional resources would be an interesting avenue for future research.

**Extending to Other Modalities.** Although our experiments focused solely on vision-language tasks, the framework should be extendable to other modalities, such as video and audio. These directions present promising topics for future exploration.

# 6 Conclusion

In this paper, we propose a novel and efficient self-improvement framework for MLLMs that does not require model-level self-feedback. We demonstrate that using our methods: 1) Significantly reduce hallucinations, enhance image-caption correspondence, and generalize to other domains, including visual reasoning, compared to the original seed model across different benchmarks. 2) Our approach enables precise control over the pair generation process, allowing us to efficiently generate preference pairs with any desired level of difference between samples. 3) We prevent cases where the positive sample is much worse than the negative one by using a lightweight CLIP model to flip samples when the score difference is too negative. Unlike traditional self-improvement methods, our approach dramatically reduces the number of parameters required during the verification process by eliminating the need for a model-level judge. Extensive experiments demonstrate that our framework effectively balances superior performance and efficiency. We hope our work inspires new strategies for managing trade-offs in the self-improvement process for MLLMs.

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

## A Ablation Studies with Different $h_{\mathbf{ratio}}$ Strategies

For a fixed $h_{\mathrm{ratio}}$ during dataset generation, we present the ablation study results in Fig. 7, which illustrate performance trends. When using a uniform distribution, we obtained the experimental results shown in Fig. 8, which illustrate variations.

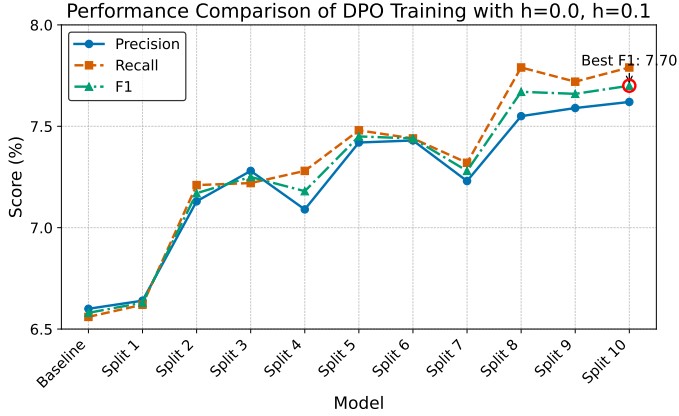

Figure 7: Performance comparison of DPO training using various CLIP score differences generated with $h_{\mathrm{ratio}} = 0.0$ and $h_{\mathrm{ratio}} = 0.1$, ranked from low to high. The best performance is highlighted.

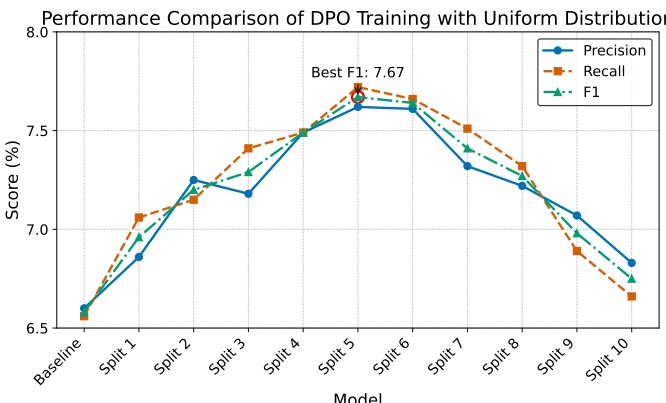

Figure 8: Performance comparison of DPO training using various CLIP score differences generated with $h_{\mathrm{ratio}}$ sampled from a uniform distribution and ranked from low to high. The best performance is highlighted.

We also conducted experiments with different combinations of the threshold and CLIP score, as shown in Fig. 9.

## B CLIP Reliability for Long Sentences

A known limitation of contrastive vision-language models like CLIP is their fixed context length (77 tokens), which can challenge the evaluation of long, descriptive captions. To address this, our framework employs a robust **atomic claim splitting** strategy similar to recent work (Jing et al., 2024; Yu et al., 2024b). This process, detailed in Algorithm 2, first splits a caption into sentences. Then, any sentence that still exceeds the token limit is broken down further into the largest possible chunks of words that fit, ensuring no information is lost and avoiding common iteration bugs. The final score for the entire caption is the average of the CLIP scores computed for each individual chunk against the image.

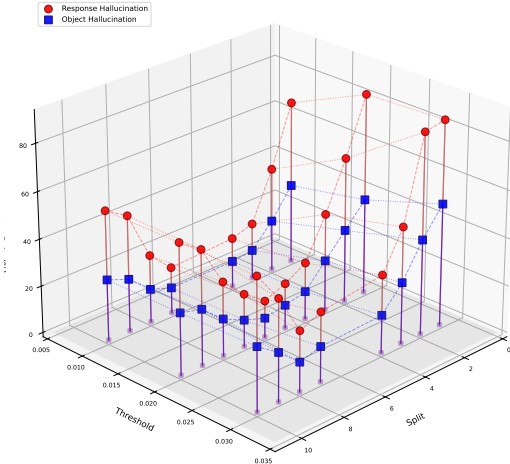

Figure 9: Comparing the performance over different combinations of a threshold and CLIP_score, with $h_{\text{ratio}}$ sampled from a Gaussian distribution. We observe that hallucination reduction depends on the CLIP_score threshold and is sensitive to larger deviations. Furthermore, after swapping, pairs with a mid-level CLIP_score difference tend to perform best, a finding that aligns with recent work (Wu et al., 2024; Deng et al., 2025).

To quantitatively validate the reliability of this approach, especially for longer captions, we conducted a rigorous analysis comparing our lightweight CLIP-based verifier against judgments from two state-of-the-art MLLMs: GPT-4o (Hurst et al., 2024) and Gemini 2.5 Pro (Comanici et al., 2025). We randomly sampled 600 preference pairs from our training data, stratified into four groups based on caption length, and measured the agreement rate on which caption in a pair was superior. The agreement between our CLIP verifier and the SOTA MLLMs was compared to a baseline agreement between the two SOTA MLLMs themselves.

The results, presented in Tab. 4, demonstrate a consistently high agreement rate that does not degrade for longer captions. Our CLIP-based verifier achieves up to **93.2%** agreement with GPT-4o, and its agreement level is comparable to the baseline agreement between GPT-4o and Gemini. This confirms that our sentence-level averaging strategy is a robust and faithful method for evaluating long captions.

Furthermore, we audited our full 100k training dataset to assess the prevalence of individual sentences exceeding the 77-token limit. We found this to be an extreme edge case, occurring in only **0.003%** of the data (3 captions). This indicates that while our sub-sentence splitting mechanism is an effective safeguard, the primary sentence-level splitting is sufficient for the vast majority of cases. These findings validate that our lightweight verification process effectively handles long captions without sacrificing scoring fidelity.

Table 4: Agreement analysis of our CLIP-based verifier against SOTA MLLMs (GPT-4o, Gemini 2.5 Pro) for captions of varying lengths. We sampled 150 preference pairs for each length category. The 'CLIP Agreement' columns show the percentage of pairs where our verifier's preference matches the MLLM's. The 'Baseline' shows the agreement between the two SOTA MLLMs. The results show our lightweight verifier's judgment is stable and highly correlated with SOTA models, even for long captions.

| Caption Length (Tokens) | Sampled Pairs (n) | GPT-4o – CLIP Agreement | Gemini – CLIP Agreement | Baseline: GPT-4o – Gemini Agreement |
|---|---|---|---|---|
| ≤77 | 150 | 83.5% | 87.8% | 84.6% |
| 78–104 | 150 | 86.8% | 89.7% | 87.9% |
| 105–132 | 150 | 93.2% | 89.7% | 92.4% |
| >132 | 150 | 87.9% | 86.9% | 83.3% |

---

**Algorithm 2** CLIP Scoring with Greedy Chunking

---

**Require:** Image $I$, Caption $C$
**Ensure:** Average CLIP Score $\bar{s}$
 1: Let $L_{max} \leftarrow 77$ {Maximum token length for the text encoder}
 2: Preprocess image: $I' \leftarrow \text{Preprocess}(I)$
 3: Split caption into sentences: $S \leftarrow \text{SentenceSplit}(C)$
 4: Initialize list of chunks: $C_{chunks} \leftarrow []$
 5: **for** each sentence $s \in S$ **do**
 6:    **if** token length of $s \leq L_{max}$ **then**
 7:       Add $s$ to $C_{chunks}$
 8:    **else**
 9:       Split sentence into words: $W \leftarrow \text{WordSplit}(s)$
10:       Initialize current chunk: $c_{curr} \leftarrow []$
11:       **for** each word $w \in W$ **do**
12:          Form a test chunk by appending the new word: $c_{test} \leftarrow \text{Join}(c_{curr}, w)$
13:          **if** token length of $c_{test} \leq L_{max}$ **then**
14:             Append $w$ to $c_{curr}$
15:          **else**
16:             Add joined $c_{curr}$ to $C_{chunks}$
17:             Start new chunk with the current word: $c_{curr} \leftarrow [w]$
18:          **end if**
19:       **end for**
20:       Add the last remaining chunk to $C_{chunks}$
21:    **end if**
22: **end for**
23: Initialize list of scores: $\mathcal{S} \leftarrow []$
24: **for** each chunk $c \in C_{chunks}$ **do**
25:    Compute similarity score: $s_c \leftarrow \text{CLIPSimilarity}(I', c)$
26:    Add $s_c$ to $\mathcal{S}$
27: **end for**
28: Compute the average score: $\bar{s} \leftarrow \text{Average}(\mathcal{S})$
29: **return** $\bar{s}$

---

## C   Unconditional Path vs. Empty String Prompt

We also conducted experiments to compare two strategies for generating negative samples: the unconditional path, which uses an instruction (e.g., "Describe image in detail") without an image input; and the empty string prompt, which uses an image with an empty string (" ") as the prompt. We analyzed responses from LLaVA-1.5-13B to both unconditional path and empty prompt inputs ($n = 10,000$ each), providing a coverage analysis in Fig. 10 and a t-SNE visualization in Fig. 11.

## D   Experimental Results on Efficiency

We included additional comparisons in terms of computational resource consumption and runtime efficiency, as shown in Tab. 5. All experiments were conducted using LLaVA-1.5-13B on a single A6000 Ada GPU. Due to limited resources, we provide estimated runtimes when full execution was not feasible. As can be seen from the results, our framework significantly reduces the resources and time needed, demonstrating greater efficiency compared to traditional self-improvement methods.

| Method | Active Parameters | Runtime | GPU Requirements |
|---|---|---|---|
| Ours | 149M | $\sim$10 minutes | 892MiB |
| RLHF/RLAIF-V | 22,000M $\sim$ 104,000M | >100 hrs | 40560MiB |

Table 5: Efficiency comparison of different methods.

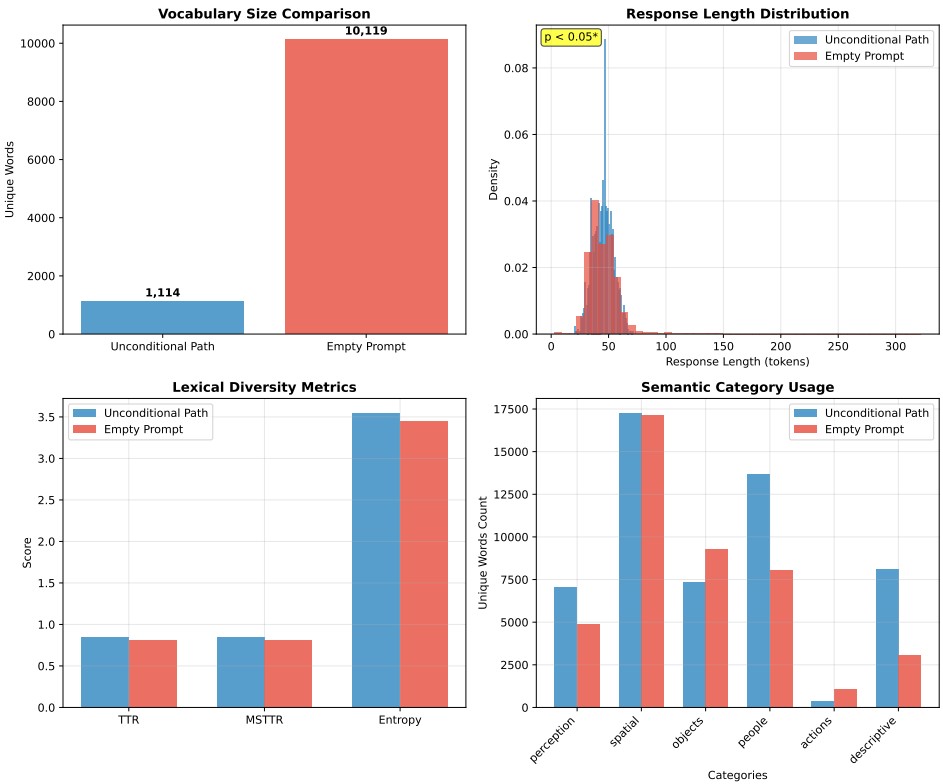

Figure 10: Coverage Analysis. (a) A vocabulary size comparison reveals a $9.1\times$ larger lexicon in empty prompt responses (10,119 vs. 1,114 unique words). (b) Response length distributions show comparable overall distributions and mean token counts (45.3 vs. 44.8 tokens, $p < 0.05$, Mann-Whitney U test). (c) Lexical diversity metrics demonstrate similarity between the two sets of responses. (d) Different distributional patterns across predefined linguistic categories.

# E Details of the IC Dataset

As mentioned in the main paper Sec 4, to comprehensively evaluate the model's performance across different caption cases, including the most challenging types, it was necessary to build a diverse dataset to address this issue.

To validate its reliability, we conducted a user study comparing human ratings of generated text with GPT-4o evaluations on the IC benchmark. We found that, on a scale from 1 to 10, the mean absolute error (MAE) between human scores and GPT-4o scores was only 1.252, demonstrating strong alignment between GPT-4o judgments and human preferences.

We provide details of each category and the number of samples collected in our IC dataset in Tab. 6.

More example visualizations of our proposed IC dataset can be found in Fig. 12.

# F Demo Examples

**Qualitative Examples Across Different Categories.** To better demonstrate the usefulness of our proposed framework, we have included additional qualitative example comparisons such as an animation image in Fig. 13, a document image in Fig. 14, and a common sense violation image in Fig. 15.

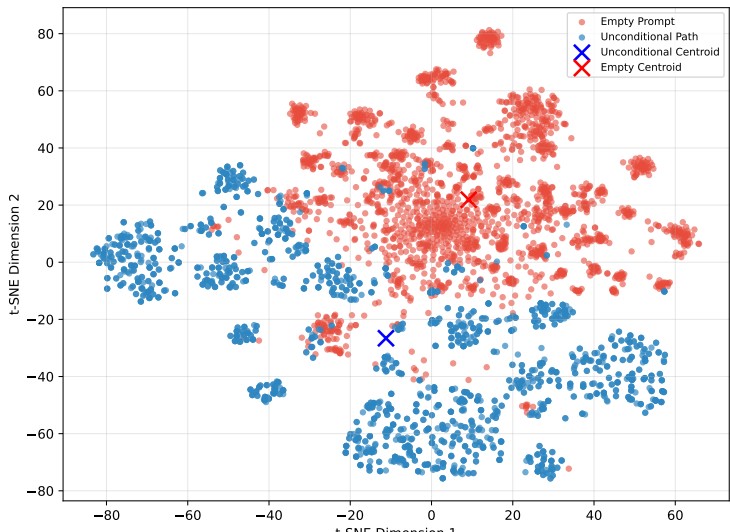

Figure 11: A t-SNE visualization of response embeddings comparing unconditional path generation (blue, $n = 2,000$) and empty prompt generation (red, $n = 2,000$) from LLaVA-1.5-13B. Points represent individual responses, projected into a 2D space from 1,000-dimensional TF-IDF features using t-SNE (perplexity=30, 1,000 iterations). The centroids for each response type are marked with crosses.

| Category | Count |
|---|---|
| abstract | 3 |
| animal | 9 |
| animation | 7 |
| artistic | 7 |
| common | 13 |
| documents | 12 |
| events | 10 |
| fashion | 9 |
| food | 9 |
| handwritten | 5 |
| illustration | 9 |
| object | 12 |
| people | 10 |
| poster | 7 |
| scenes | 9 |
| technology | 9 |
| vehicle | 10 |
| **Total** | **150** |

Table 6: Category and image counts of our IC dataset.

**GPT-4o evaluation.** For the GPT-4o evaluation, each caption was processed by GPT-4o to separately generate precision analysis and recall analysis. The precision analysis was used to compute the precision score, and the recall analysis was used to compute the recall score. The detailed prompts are shown in Fig. 16.

Detailed examples of precision analysis and scores are provided in Fig. 17, and examples of recall analysis and scores are shown in Fig. 18.

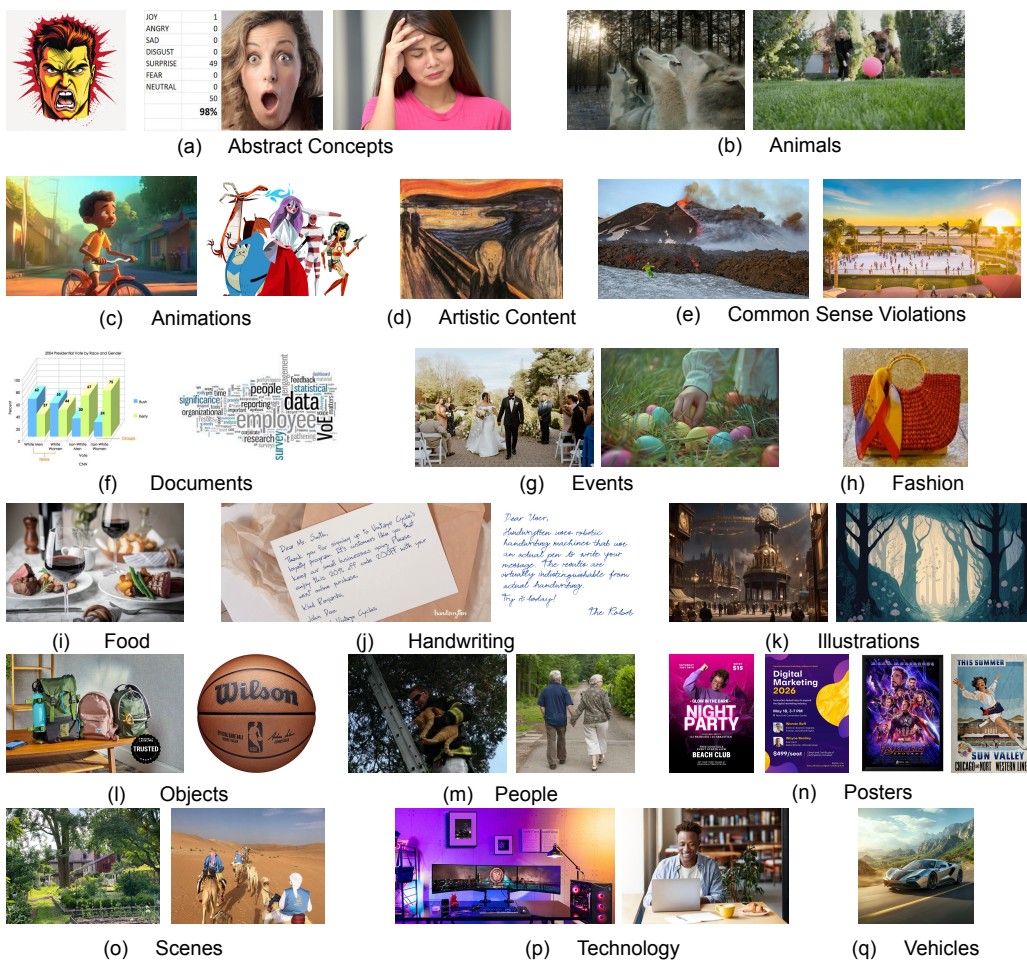

Figure 12: Examples from the IC dataset. The dataset covers a wide range of diverse categories important for image captioning tasks, including abstract concepts, animals, animations, artistic content, common sense violations, documents, events, fashion, food, handwriting, illustrations, objects, people, posters, scenes, technology, and vehicles.

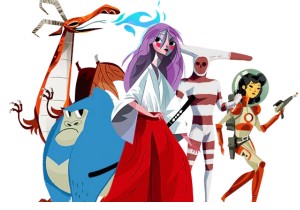

The image features a colorful and vibrant scene with a woman in the center, surrounded by various characters. She is wearing a white dress and holding a sword, possibly a ninja sword. The woman appears to be the main focus of the scene, with her purple hair standing out.

There are several other characters in the image, including a dragon on the left side and a monkey on the right side. The monkey is holding a guitar, adding a unique element to the scene. The combination of the woman, the sword, and the diverse characters creates a lively and engaging visual experience.

LLaVA 1.5 13b

The image features a colorful and vibrant anime-style artwork with various characters. The main focus is a girl dressed in a kimono, standing in the center of the scene, surrounded by other characters. The girl is holding a sword, and her outfit is adorned with a belt.

In addition to the girl, there are several other characters in the image, including a blue monster and a dragon, both of which are positioned on the left side of the scene. The artwork showcases the diverse and lively nature of the characters, creating a captivating and engaging visual experience.

Ours

Figure 13: Qualitative comparison of animation image captions. The left panel shows the description generated by LLaVA-1.5-13B, while the right panel presents our method's output. Both descriptions capture the vibrant, anime-style scene featuring diverse characters, but our method demonstrates superior accuracy and specificity in detailing the central figure (a girl in a kimono) and surrounding elements (a blue monster and a dragon on the left). This highlights our approach's improved understanding of nuanced visual elements and context, enabling richer and more precise visual descriptions.

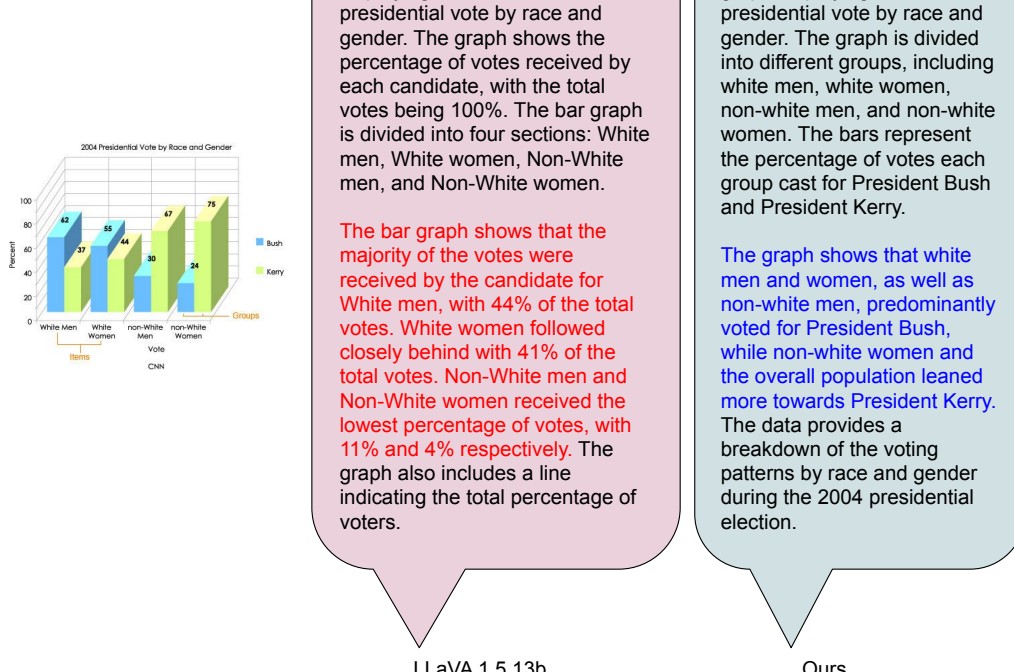

The image is a bar graph displaying the results of the 2004 presidential vote by race and gender. The graph shows the percentage of votes received by each candidate, with the total votes being 100%. The bar graph is divided into four sections: White men, White women, Non-White men, and Non-White women.

The bar graph shows that the majority of the votes were received by the candidate for White men, with 44% of the total votes. White women followed closely behind with 41% of the total votes. Non-White men and Non-White women received the lowest percentage of votes, with 11% and 4% respectively. The graph also includes a line indicating the total percentage of voters.

LLaVA 1.5 13b

The image features a bar graph displaying the 2004 presidential vote by race and gender. The graph is divided into different groups, including white men, white women, non-white men, and non-white women. The bars represent the percentage of votes each group cast for President Bush and President Kerry.

The graph shows that white men and women, as well as non-white men, predominantly voted for President Bush, while non-white women and the overall population leaned more towards President Kerry. The data provides a breakdown of the voting patterns by race and gender during the 2004 presidential election.

Ours

Figure 14: Qualitative comparison of document image captions. The left panel contains the description generated by LLaVA-1.5-13B, while the right panel shows the output from our method. Both descriptions analyze a bar graph depicting the 2004 U.S. presidential vote by race and gender, but our method provides a more concise and focused interpretation. It highlights key trends such as voting patterns for Bush and Kerry across demographic groups, emphasizing accuracy and clarity in extracting meaningful insights from visualized data.

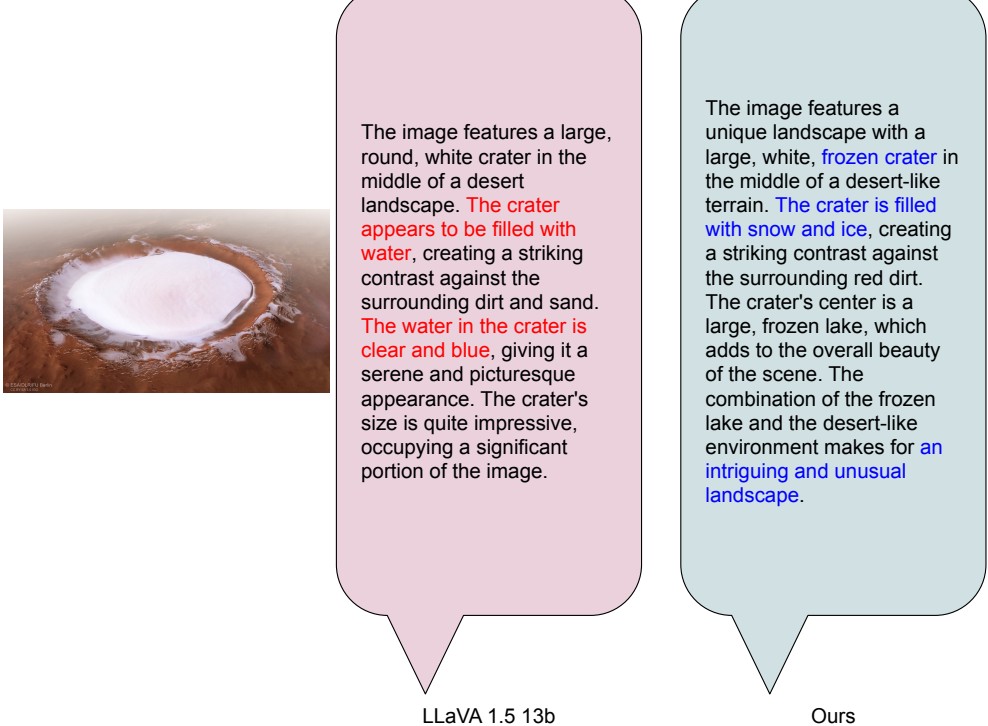

Figure 15: Qualitative comparison of common sense violation image captions. The left panel shows the description generated by LLaVA-1.5-13B, while the right panel presents our method's interpretation. While both descriptions recognize the unique setting of a white crater within a desert-like terrain, our method provides a more accurate depiction by identifying the crater as frozen and filled with snow and ice, rather than water. This enhanced understanding highlights our model's ability to handle complex and counterintuitive visual elements, ensuring clarity and correctness in scenarios that defy common expectations.

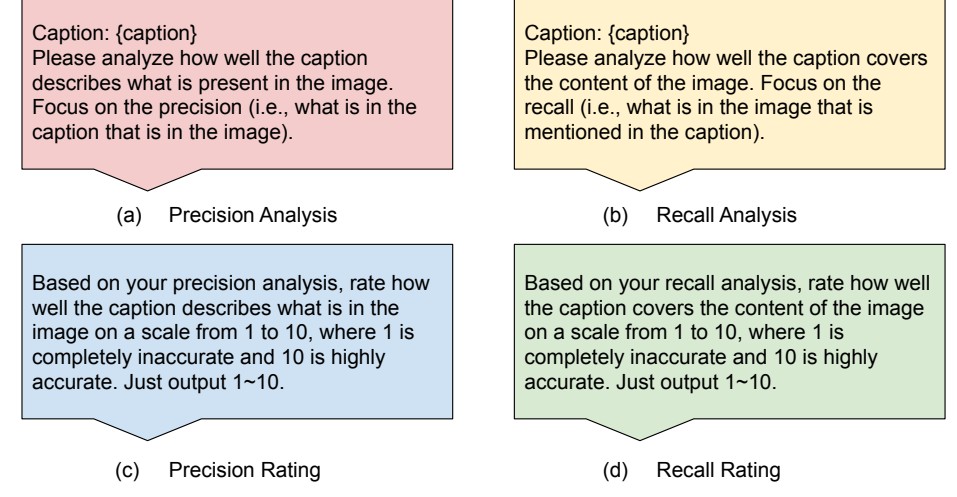

Figure 16: Prompts used for GPT-4o evaluation. We first use (a) and (b) to obtain precision and recall analysis separately, and then, combining these analyses, we use (c) and (d) to calculate the final precision and recall scores accordingly.

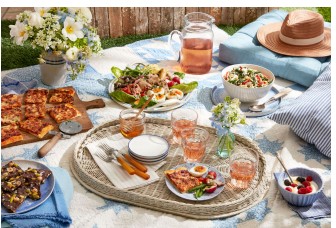

The image features a picnic setting with a blanket spread out on the grass. The blanket is adorned with a variety of food and drinks, creating a delightful and inviting atmosphere. There are several plates of food, including pizza and salad, as well as a bowl of fruit.

In addition to the food, there are multiple cups and a wine glass placed on the blanket, offering a selection of beverages for the guests. A vase is also present, adding a decorative touch to the scene.

Utensils such as forks, knives, and spoons are laid out on the blanket, ready for use. A potted plant can be seen nearby, further enhancing the outdoor ambiance. A chair is positioned close to the blanket, providing a comfortable seating option for those enjoying the picnic.

The caption provides a general overview of the scene and captures many elements present in the image, but there are some inaccuracies and omissions. Here's an analysis of precision:

1. **Setting**: The caption accurately describes the picnic setting with a blanket on the grass, which is indeed depicted in the image.

2. **Food Items**:
- The mention of pizza aligns with the visible pizza on the wooden board.
- The salad is correctly noted as being part of the spread, but the caption could be more specific about the contents of the salad.
- The mention of a bowl of fruit is somewhat misleading; while there are snacks that resemble fruit (the small dish has fruit), there isn't a clear bowl of only fruit.

3. **Beverages**:
- The caption correctly states the presence of multiple cups. However, it inaccurately refers to a wine glass; the image shows glasses of what appears to be a juice or iced tea rather than wine glasses.

4. **Decorative Elements**:
- The vase with flowers is accurately referenced, adding a decorative touch.
- The mention of a potted plant is inaccurate; while there is foliage in the background, it cannot be clearly identified as a potted plant.

5. **Utensils**:
- The caption mentions utensils such as forks, knives, and spoons, which aligns with what is laid out on the blanket, but it specifies fewer types of utensils than actually present. The scene includes forks but not knives.

6. **Seating Description**:
- The caption states that a chair is positioned close to the blanket, which is not visible in the image; instead, there are cushions, not a chair.

Overall, the caption captures the general essence of the picnic scene, but it contains inaccuracies regarding the specific food items, the types of drinks, the presence of a chair, and details about the utensils. More precise descriptions and accuracy in what is actually visible would improve the caption significantly.

6

(a)   Image with caption

(b)   Precision analysis and score

Figure 17: GPT-4o analyzing the precision of a caption given a food image and rating the score.



The image features a piece of notebook paper with a handwritten essay on it. The essay is about the ancient Greeks and their mathematical contributions. The paper contains a diagram drawn in blue ink, which is likely related to the essay's content. The essay is written in cursive, giving it a personal and historical feel. The combination of the handwritten text and the diagram suggests that the essay is a detailed exploration of the ancient Greeks' mathematical achievements.

The caption provides a solid overview of the content of the image, focusing on several key elements:

1. **Notebook Paper and Handwritten Essay**: The caption correctly notes that the image depicts a piece of notebook paper with a handwritten essay, which matches the visual content.

2. **Topic of the Essay**: It accurately identifies that the essay concerns the ancient Greeks and their mathematical contributions, which is a central theme in the text shown in the image.

3. **Diagram in Blue Ink**: The caption mentions a diagram drawn in blue ink, which aligns with what is visible in the image. It connects this diagram to the essay's content, although it does not specify what the diagram illustrates.

4. **Cursive Writing**: The caption highlights that the essay is written in cursive, reflecting the personal touch and historical context, which corresponds to the style seen in the image.

5. **Detailed Exploration**: The assessment of the essay as a detailed exploration of the ancient Greeks' mathematical achievements aligns with the informative nature of the text present in the image.

Overall, the caption effectively captures the primary elements of the image, including the handwriting, topic, and visual characteristics. However, it could enhance its precision by briefly mentioning the content of the diagram to deepen the connection between the textual and visual components.

9

(a)   Image with caption

(b)   Recall analysis and score

Figure 18: GPT-4o analyzing the recall of a caption given a handwritten image and rating the score.

