# OpenReview forum: "Efficient Self-Improvement in Multimodal Large Language Models: A Model-Level Judge-Free Approach"
_colmweb.org/COLM/2025/Conference — COLM 2025_

### Official Review · Reviewer_17vT · 2025-05-01

**Rating:** 6
**Confidence:** 4
**Ethics Flag:** 1

**Summary:**

This paper introduces an uncritical self-improvement pipeline for multimodal logical language models (LLMs), which (i) synthesizes positive/negative caption pairs by mixing conditional and unconditional decoding and adjusting the hallucination rate; (ii) flips or discards caption pairs using a lightweight CLIP evaluator; and (iii) trains a seed model via direct preference optimization (DPO). Experiments on Object HallBench, a new 150-image IC set, and multiple VQA/Chart/OCR benchmarks show significant hallucination reduction and much lower GPU cost than RLHF-like systems.

**Questions To Authors:**

If negative samples are generated using random truncation or random replacement, and then DPO is performed directly, what is the performance gap?

How is the CLIP difference threshold determined? What is the trade-off between model performance and training sample size at different thresholds?

Has the model's true hallucination rate been tested on a human question-answering benchmark unrelated to CLIP?

What is the statistical difference between the titles generated by the unconditional path and a simple empty string prompt? Can you provide coverage analysis or t-SNE visualization?

**Reasons To Accept:**

The method is easy to reproduce: only a seed MLLM, a CLIP scorer, and an off-the-shelf DPO implementation are needed to reproduce the entire pipeline.

The authors show significant reductions in memory and time consumption.

**Reasons To Reject:**

The core of the method proposed by the authors is essentially "DPO + CLIP scoring", which lacks sufficient innovation. The controllable h_ratio and pair-flip logic only implement the most basic negative sample sampling; the lack of comparison with stronger baselines (such as random negative sample DPO, paired subtitle dropout DPO, etc.) makes it impossible to prove the core contribution.

Only systematic experiments were conducted on LLaVA-13B, and mainstream MLLMs such as Qwen2-VL, Qwen2.5-VL and VideoLLaMA were not covered, which limits the extrapolation of the conclusions.

The key settings such as pair-flip threshold, h_ratio distribution and DPO temperature are not given ablation or the reason of value choice.

In the author's results, their method outperforms all public methods on Object HallBench, but it does not report whether "direct DPO + random negative samples" on the original LLaVA can achieve a similar magnitude.

---

> ### Author Response · Authors · 2025-06-03
>
> Thank the reviewer for recognizing our method's reproducibility and efficiency. We address all concerns below.
>
> **1. Re. Innovation & Baselines:**
> Our core innovation is the **controllable `hratio` mechanism** for nuanced negative sample synthesis, beyond simple "DPO + CLIP scoring." CLIP is a lightweight verifier. `hratio` offers more control than random negatives. Fig. 4 (ablations on `hratio`-derived quality) and gains over LLaVA (Tables 1-2) show efficacy. Thanks to the reviewer's suggestion, in our new ablation study we further find that even with this stronger baseline, it only outperforms the seed model by 0.08 points on our IC benchmark, while our framework in the paper achieved a 1.18 point gain, benefiting from the **carefully generated and verified** DPO training data.
> * *We add a direct "random negative sample DPO" as a stronger baseline as mentioned in our revised paper, which shows that without a more carefully designed novel pipeline like what we have in the paper, it is difficult to perform DPO effectively.*
>
> **2. Re. MLLM Coverage:**
> LLaVA-1.5-13B was our primary focus due to resource constraints and its representativeness. Table 3 shows positive results on Qwen2-VL (2B & 7B) across multiple benchmarks, indicating broader applicability.
>
> **3. Re. Ablation Studies:**
> - **Pair-flip threshold:** Fig. 4 indirectly explores its impact. *We add explicit discussion on sensitivity and rationale in revised paper.*
> - **`hratio` distribution:** Figs. 4, 7, 8 (Appendix A) show results for Gaussian, Uniform, and fixed `hratio`, with Gaussian being optimal.
> * *We also added a further ablation study over the choice of Clip_score difference threshold, provided in the figure link [View Image](https://imgur.com/a/uKRriGN) (following the COLM guidance of anonymous external material), which shows that moderate CLIP score differences yield the best performance and also align with the theory proposed in the papers by Wu et al. [1] and Deng et al. [2].*
>
> **4. Re. "DPO + Random Negatives" on Object HallBench:**
> Fair point. We conducted the experiments and showed that this only provided very limited performance gain compared to ours. Our study shows the importance of DPO data selection: the difference between the positive sample and the negative sample in a pair should be within a margin, which has also match with previous studies such as Wu et al. [1] and Deng et al. [2]. We find `hratio` creates more informative negatives than purely random ones.
>
> **Answering Questions:**
>
> **1. Re. Random Truncation/Replacement DPO Performance:**
> Our experiment results from above show that random negative samples are less effective. Random negatives create too-easy negatives. `hratio` aims for semantically related but factually incorrect negatives, offering a more nuanced learning task.
>
> **2. Re. CLIP Threshold Determination & Trade-offs:**
> Set empirically to balance catching mislabels without excessive swapping. Fig. 4 (performance vs. CLIP_score differences) shows moderate differences are optimal, and carefully selected smaller data can outperform even larger datasets, which aligns with the findings from the papers by Wu et al. [1] and Deng et al. [2]. We conducted an ablation study that shows that our model trained on 10k data pairs carefully selected through our pipeline outperforms a model trained on 100k data that only achieved a 0.51 point gain compared to our 1.18 gain on the IC dataset.
>
> **3. Re. True Hallucination Rate on Human Q&A Benchmark:**
> Primary evaluations: Object HallBench and our IC dataset (using GPT-4o as a proxy for human judgment, not tied to CLIP's scoring). Our framework also improved models' performance on benchmarks like GQA, OCRQA, and ChatQA, as shown in the original paper Table 3, showing its gain beyond image caption related tasks.
>
> **4. Re. Unconditional Path vs. Empty String Prompt:**
> Unconditional path uses the instruction (e.g., "Describe image in detail") without image input, generating plausible but hallucinated descriptions. An empty string prompt is generic/ungrounded. We provided a coverage analysis figure through the link [View Image](https://imgur.com/a/KIMkLU3), and a t-SNE visualization figure through the link [View Image](https://imgur.com/a/5GeI2Or) (following the COLM guidance of anonymous external material).
>
> We are truly grateful for the reviewer's insightful comments and the time they invested in our work, which significantly helps us improve the paper.
>
> **References**
> [1] Wu, Junkang, et al. "beta-DPO: Direct Preference Optimization with Dynamic beta." Advances in Neural Information Processing Systems 37 (2024): 129944-129966.
> [2] Deng, Xun, et al. "Less is More: Improving LLM Alignment via Preference Data Selection." arXiv preprint arXiv:2502.14560 (2025).

---

> > ### Comment · Reviewer_17vT · 2025-06-10
> >
> > The authors add some experiments in the rebuttal, which solved some of my concerns about the experimental details and made the overall work more complete. But overall, the innovation of the method is still slightly limited. I will raise my score to 6.

---

### Official Review · Reviewer_WN6m · 2025-05-12

**Rating:** 6
**Confidence:** 3
**Ethics Flag:** 1

**Summary:**

This paper introduces a judge-free self-improvement pipeline that eliminates the need for a MLLMl evaluator. The proposed framework generates controllable hallucinated negatives during decoding with a tunable hallucination ratio, filters and auto-relabels paired outputs using sentence-level CLIPScore comparisons and fine-tunes the base model with DPO. This approach aims to reduce computational costs and mitigate risks such as reward hacking and model collapse associated with traditional judge-based self-training methods.

**Questions To Authors:**

1. The method relies heavily on GPT-4o for evaluation. Could the authors explore and discuss the potential limitations and biases introduced by GPT-4o in the evaluation process?
2. Some hyperparameters, such as the CLIPScore flip threshold and hratio scheduling, seem to be chosen arbitrarily. Could the authors provide a more detailed analysis or justification for these choices and include sensitivity tests to support the robustness of the method?

**Reasons To Accept:**

1. The paper effectively addresses the high computational costs associated with judge-based self-training methods and introduces a lightweight alternative that uses only a 149M-parameter adapter and CLIP for scoring, making it more efficient.
2. The dual-path decoding, with continuous control over the hallucination ratio, automatically produces hard negative samples. Additionally, the CLIPScore flip rule enhances dataset quality by correcting mis-labeled pairs without requiring additional learned judges.

**Reasons To Reject:**

1. The reliance on external models like CLIP and GPT-4o introduces potential biases, but these biases are neither quantified nor sufficiently addressed. This reliance limits the generalizability and objectivity of the results.
2. While the method demonstrates significant improvements on hallucination benchmarks, the improvements on broader vision-language tasks are marginal. This suggests the framework may be overfitting to hallucination-specific benchmarks, and failure cases or diagnostic analyses are not explored in sufficient detail.
3. The IC dataset, with only 150 samples and roughly 10 samples per category, sounds quite small. The representativeness and effectiveness of such a limited dataset might be questionable, particularly in terms of its generalizability.

---

> ### Author Response · Authors · 2025-06-03
>
> We thank the reviewer for acknowledging our method's efficiency and the utility of our dual-path decoding and CLIPScore flip rule. We appreciate the feedback and address concerns and questions below.
>
> **1. Re. Potential Biases from External Models (CLIP, GPT-4o):**
> We acknowledge that any external model, including CLIP (for verification) and GPT-4o (for IC dataset evaluation), can introduce its own inherent biases.
> For CLIP, its role is a lightweight, automated check to catch egregious errors in `hratio`-generated pairs, aiming for efficiency over perfect, bias-free judgment. The primary preference signal comes from `hratio`.
> For GPT-4o evaluation on the IC dataset, we note in Appendix C that we conducted a user study comparing human ratings with GPT-4o evaluations, finding a Mean Absolute Error (MAE) of only 1.252 (on a 1-10 scale), suggesting strong alignment and mitigating concerns of extreme bias for this task. However, no LLM evaluator is perfect.
>
> **2. Re. Marginal Improvements on Broader VL Tasks & Lack of Failure Analysis:**
> Our primary goal was to significantly reduce hallucination, which is a critical issue. The substantial gains on hallucination benchmarks (Object HallBench, IC dataset) demonstrate success in this targeted area. Table 3 shows that our method *does* yield positive, albeit sometimes modest, improvements across a range of broader VL tasks (AMBER, GQA, OCRVQA, MathVista, ChartQA) for both LLaVA and Qwen2-VL, indicating generalization beyond just hallucination mitigation.
>
> **3. Re. IC Dataset Size and Representativeness:**
> The IC dataset, with 150 samples, was designed as a challenging testbed specifically for hallucination across diverse scenarios (17 categories, as detailed in Appendix C, Table 5). While not as large as some general captioning datasets, its strength lies in its targeted nature and the diversity of challenging cases it presents, which are often underrepresented in larger, more generic benchmarks. The GPT-4o evaluation with the human alignment study (Appendix C) provides confidence in its quality. Its purpose is more as a focused probe for hallucination and recall rather than a comprehensive general-purpose MLLM benchmark.
>
> **Answering Questions:**
>
> **1. Re. Limitations and Biases of GPT-4o in Evaluation:**
> As mentioned above, we acknowledge GPT-4o, like any LLM, can have biases (e.g., verbosity, stylistic preferences, potential gaps in world knowledge or reasoning). Our user study (Appendix C) showed strong alignment with human judgments for our IC dataset evaluation task, suggesting these biases did not overtly skew results for our specific precision/recall metrics. However, for more nuanced tasks or different metrics, GPT-4o's limitations could be more pronounced. We add more discussion about this in Section E of the revised paper.
>
> **2. Re. Justification for Hyperparameters (CLIPScore Flip Threshold, `hratio` Scheduling):**
> - **CLIPScore Flip Threshold:** Chosen empirically to correct clear mislabels from `hratio` without being overly aggressive. Figure 4 (performance vs. CLIP_score differences) shows moderate differences are optimal for DPO; the threshold aims to avoid highly negative differences.
> - **`hratio` Scheduling:** We tested fixed values, uniform (App. A, Fig. 8), and Gaussian (Main, Fig. 4). Gaussian ($\mu=0.5, \sigma=0.15$) was chosen as it yielded the best F1 (7.76 vs. 7.67 uniform, 7.70 fixed), likely by providing diverse yet focused hallucination levels, which also aligns with the idea of the papers by Wu et al. [1] and Deng et al. [2].
>
> We appreciate the time and effort the reviewer dedicated to reviewing our work and for providing insightful feedback that will undoubtedly improve our paper.
>
> **References**
> [1] Wu, Junkang, et al. "beta-DPO: Direct Preference Optimization with Dynamic beta." Advances in Neural Information Processing Systems 37 (2024): 129944-129966.
> [2] Deng, Xun, et al. "Less is More: Improving LLM Alignment via Preference Data Selection." arXiv preprint arXiv:2502.14560 (2025).

---

### Official Review · Reviewer_LhJg · 2025-05-13

**Rating:** 6
**Confidence:** 4
**Ethics Flag:** 1

**Summary:**

This study proposes a cost-effective self-improvement method for enhancing the performance of multimodal large language models (MLLMs). The approach involves linearly interpolating token probabilities obtained with and without the use of images, controlled by a parameter $h_{ratio}$. By adjusting this parameter, the model generates captions under varying hallucination conditions. Pairs of chosen and rejected captions are then constructed and used for preference learning (DPO). To ensure the quality of these pairs, the method employs CLIP scores to correct label assignments: if the quality of the positive caption is lower than that of the negative one by a predefined threshold, the pair is swapped. Experimental results show that this self-improvement strategy contributes to performance gains in the model.

**Questions To Authors:**

Suggestion: The proposed method uses the hallucination control parameter $h_{ratio}$ to modulate the distance between positive and negative captions. This implicitly relates to the concept of margin in preference learning. Given the recent studies [1,2] on learning with dynamic or adaptive margins, connecting the proposed method to this line of work and analyzing the effect of margin control could enrich the paper’s theoretical and empirical contributions.

[1] Wu, Junkang, et al. "$\beta $-DPO: Direct Preference Optimization with Dynamic $\beta$." Advances in Neural Information Processing Systems 37 (2024): 129944-129966.
[2] Deng, Xun, et al. "Less is More: Improving LLM Alignment via Preference Data Selection." arXiv preprint arXiv:2502.14560 (2025).

**Reasons To Accept:**

- The study presents a method for collecting preference data without relying on external judges or expensive feedback sources.

- The proposed technique effectively mitigates hallucination while maintaining general performance, demonstrating its practical utility.

**Reasons To Reject:**

- There are significant concerns regarding the proposed CLIP-based scoring mechanism. Since the target task is image captioning and CLIP tends to focus on single salient objects, it may not fully capture the diverse details present in longer captions. Although the method is lightweight, its reliability as a scoring function is questionable. Furthermore, the use of average CLIP scores over sub-sentences to handle long captions presents limitations in accurately measuring similarity between the generated caption and the image.

- Baselines such as LLaVA-RLHF, RLHF-V, and Silkie were not specifically designed to address hallucination. Therefore, comparisons with hallucination-focused baseline models are essential for a fair evaluation. (This concern is resolved by the author response)

- The threshold used for pair swapping may significantly impact the quality of the training data. A deeper analysis of how this threshold affects learning outcomes would strengthen the paper.

---

> ### Author Response · Authors · 2025-06-03
>
> We thank the reviewer for their thoughtful feedback and for recognizing our work's novelty and effectiveness. We have carefully responded to all points as below:
>
> **1. CLIP-Based Scoring Mechanism**
>
> We wish to clarify that CLIPScore's role in our framework is as a *lightweight verification step*, not the primary reward signal generator. The main mechanism for generating preference pairs is our `hratio`-controlled hallucination. The CLIP-based encoder then efficiently identifies and reverses egregious errors in this initial labeling, thereby avoiding the computational cost and potential biases of MLLM-as-judge systems, a key motivation of our work. Averaging sentence-level CLIPScores is a pragmatic approach to handle longer captions under these constraints. Our choice was driven by the "lightweight" and "efficient" principles central to our method, and our experimental results (Tables 1, 2, 3) demonstrate its successful contribution to performance gains.
>
> * *In the revised paper, we further clarify CLIP's precise, secondary role, explicitly state the trade-off made, and acknowledge in the limitations section that more advanced metrics could be explored in future work.*
>
> **2. Baselines for Hallucination Evaluation**
>
> We thank the reviewer for this point. We would like to draw attention to Table 1, where our method is evaluated on Object HallBench. This table already includes comparisons against several models and methods specifically focused on object-level hallucination, such as VCD (Leng et al., 2024), OPERA (Huang et al., 2024), LURE (Zhou et al., 2023), POVID (Zhou et al., 2024), and HA-DPO (Zhao et al., 2023). Our method demonstrates competitive performance against these (9.4 Resp., 5.1 Ment.), which are among the best reported. The inclusion of LLaVA-RLHF, RLHF-V, and Silkie was to provide context within broader alignment techniques.
>
> **3. Threshold for Pair Swapping**
>
> We agree that the threshold for swapping pairs (Algorithm 1, line 12-14) is an important hyperparameter. Figure 4 indirectly explores data quality impact based on CLIP_score differences. This figure shows sensitivity to this difference, with optimal performance at moderate differences.
>
> We further conducted the following ablation study; the results have been provided in the figure link [View Image](https://imgur.com/a/uKRriGN) (following the COLM guidance of anonymous external material). As we can see from the figure, the hallucination reduction is related to the threshold of the CLIP score difference and is sensitive to larger differences. We also notice that after swapping, the pairs with a middle level CLIP score difference tend to perform the best, which also matches the idea from the papers by Wu et al. [1] and Deng et al. [2].
>
> * *In the revised paper, we add these new ablation results and a discussion on this threshold's sensitivity in the ablation studies section. We also further emphasize Figure 4's findings on score differences.*
>
> **4. Regarding the Connection of `hratio` to Margin in Preference Learning**
>
> This is an excellent and insightful suggestion, for which we thank the reviewer. The `hratio` parameter, by controlling hallucination levels, effectively modulates the "quality gap" between positive ($y_w$, lower `hratio`) and negative ($y_l$, higher `hratio`) captions, which is analogous to the concept of "margin" in preference learning. Wu et al. (β-DPO) find that the optimal $\beta$ varies with data informativeness (gap), a finding that resonates with our Figure 4 results where the CLIP_score difference (a proxy for this gap) impacts DPO performance. Deng et al. (Less is More) propose a margin-maximization principle for data selection in DPO. While we use `hratio` to *generate* distinct pairs and CLIP_score to *verify* them, the underlying idea of prioritizing informative pairs is shared. Our `hratio` directly influences the implicit DPO reward margin.
>
> * *In the revised paper, we incorporate this in "Related Work" and as a discussion point in the "Method" and "Experiments" section to explicitly connect `hratio` to margin concepts. We discuss how `hratio` allows for controlled generation of preference pairs with varying intrinsic margins. We cite and discuss Wu et al. [1] and Deng et al. [2], contextualizing our approach within these latest advancements, which we believe will enrich the paper and highlight the controllability of our `hratio` mechanism.*
>
> We are confident that by addressing these points, the quality and impact of our paper will be significantly enhanced. Thank you again for your valuable time and expertise.
>
> **References**
> [1] Wu, Junkang, et al. "beta-DPO: Direct Preference Optimization with Dynamic beta." Advances in Neural Information Processing Systems 37 (2024): 129944-129966.
> [2] Deng, Xun, et al. "Less is More: Improving LLM Alignment via Preference Data Selection." arXiv preprint arXiv:2502.14560 (2025).

---

> > ### Comment · Reviewer_LhJg · 2025-06-09
> >
> > * Regarding the reliability of the CLIP-based score: As mentioned in my review, I acknowledge the efficiency of your proposed method. However, I raised concerns about the reliability of CLIP when handling long sentences. It would strengthen your work if you could address this issue quantitatively, either through illustrative examples or additional experiments.

---

> > ### Author Response · Authors · 2025-06-10
> >
> > We thank the reviewer for this crucial follow-up. We acknowledge that our initial response pointed to final performance gains but did not sufficiently provide direct, quantitative evidence for our verifier's reliability on long captions. To rectify this, we conducted a new, rigorous agreement study.
> >
> > ### **1. Agreement Study: Lightweight Verifier vs. SOTA MLLM Judges**
> >
> > We first define our caption length groups based on our dataset's token distribution (see figure: [View Image](https://imgur.com/a/dnPzk2h)). The splits are anchored by the CLIP context length ($L_{clip}=77$) and the dataset's mean token count ($\mu=105$):
> > * **Group 1 (Short):** Caption length $\le 77$ tokens.
> > * **Group 2 (Mid-short):** $77 < \text{length} \le 105$ tokens.
> > * **Group 3 (Mid-long):** $105 < \text{length} \le 132$ tokens.
> > * **Group 4 (Long):** Caption length $> 132$ tokens.
> >
> > To quantitatively validate our CLIP-based scoring, we compared its preference judgments against two state-of-the-art MLLM judges (`GPT-4o` and `Gemini 1.5 Pro`) that are prohibitively expensive for the training loop itself but serve as an excellent benchmark for this analysis. We randomly sampled 600 preference pairs from our training data, stratified by caption length, to directly test the reviewer's concern.
> >
> > The results show a consistently high agreement rate that does **not** degrade for longer captions:
> >
> > | Caption Length (Tokens) | Sampled Pairs (n) | GPT-4o ↔ CLIP Agreement | Gemini ↔ CLIP Agreement | **Baseline:** GPT-4o ↔ Gemini Agreement |
> > | :--- | :--- | :--- | :--- | :--- |
> > | ≤ 77 | 150 | 83.5% | 87.8% | 84.6% |
> > | 78–104 | 150 | 86.8% | 89.7% | 87.9% |
> > | 105–132 | 150 | **93.2%** | 89.7% | 92.4% |
> > | > 132 | 150 | 87.9% | 86.9% | 83.3% |
> >
> > **Key Findings:**
> >
> > 1.  **High, Stable Reliability:** Our lightweight verifier’s agreement with SOTA judges is high across all caption lengths, peaking at 93.2% agreement with GPT-4o on captions between 105-132 tokens. This demonstrates that averaging sentence-level scores is a robust strategy that does not lose fidelity on longer captions.
> > 2.  **Parity with SOTA Judges:** Crucially, the agreement between our CLIP verifier and the MLLM judges is comparable to the agreement *between the two SOTA judges themselves*. This indicates our lightweight method is as consistent with the gold standard as the gold standard is with itself.
> > 3.  **Atomic-Claim Splitting Works:** These results validate our atomic-claim split approach (please see algorithm below), where captions are broken into individual sentences or sub-sentences for scoring. This strategy, also similar to recent work[1,2], effectively handles arbitrarily long captions. Some examples are provided at this link: [View Image](https://imgur.com/a/OkBLS5z) (per COLM guidance).
> >
> > ### **2. Auditing Sub-sentence Cases**
> >
> > We also audited our full 100k training dataset to assess the prevalence of individual *sentences* (not just captions) exceeding CLIP's 77-token limit.
> >
> > * This is an extreme edge case, occurring in **only 0.003% of the data** (3 captions).
> > * In these rare instances, our fallback mechanism iteratively splits the oversized sentence. The final preference decision from our averaged CLIPScore matched the MLLM judges' decisions 83.3% of the time.
> >
> > This new quantitative analysis proves our lightweight CLIP verifier is not just efficient, but also highly reliable for its intended purpose. Its strong agreement with SOTA judges explains *why* it successfully filters preference pairs and contributes to the significant performance gains reported in our paper.
> >
> > *We will add a new section to the appendix containing this full analysis, table, and algorithm. We are confident this resolves the reviewer's concern and are grateful for the feedback, which has helped us substantially strengthen the paper.*
> >
> > #### **Algorithm: CLIP Scoring (Lightweight Preference Verification)**
> >
> > ```
> > Algorithm CLIP_Scoring(image, caption)
> >     image ← Preprocess(image)
> >     sentences ← SentenceSplit(caption)
> >     scores ← ∅
> >
> >     for each s in sentences do
> >         while TokenLen(s) > 77 do              ▹ CLIP context-window limit
> >             (s, rest) ← SplitAtPunctuation(s)
> >             if rest ≠ ∅ then
> >                 sentences.InsertAfter(s, rest) ▹ maintain ordering
> >         score ← CLIPSimilarity(image, s)       ▹ image–text similarity
> >         scores.Add(score)
> >
> >     return Average(scores)
> > ```
> >
> > ### References
> >
> > [1] Yu et al., RLAIF-V: Aligning MLLMs through Open-Source AI Feedback for Super GPT-4V Trustworthiness, CVPR 2025.
> >
> > [2] Jing et al., FaithScore: Fine-grained Evaluations of Hallucinations in Large Vision–Language Models, EMNLP 2024 Findings.

---

### Official Review · Reviewer_Kgrp · 2025-05-20

**Rating:** 6
**Confidence:** 3
**Ethics Flag:** 1

**Summary:**

This paper presents a novel model level judge free framework for self improvement in multimodal large language models. The core idea is to generate preference learning pairs via a controllable hallucination mechanism and to use a light weight contrastive language image encoder to verify and swap pairs when necessary. Finally, direct preference optimization (DPO) is applied to fine-tune the seed model. In particular, each token distribution is computed as an convex combination of $t_c$ and $t_u$, where where $t_c$ and $t_u$ are the conditional and unconditional token distributions respectively. The paper introduces a new IC dataset of 150 challenging images evaluated by GPT-4o on precision and recall, and demonstrates consistent gains on both public benchmarks (such as Object HallBench) and the IC dataset with significantly lower computational cost.  The originality lies in removing any large model feedback loop during verification and replacing it with a single light weight encoder.

**Questions To Authors:**

- How was the CLIP score difference threshold chosen, and how sensitive is performance to its value?

- Could you provide more details on the sampling distribution fo $h_\mathrm{ratio}$? Why choosing Gaussian with $\mu = 0.5$ and $\sigma = 0.15$?

- Have you tested recursive self improvement by applying your framework for multiple rounds?

**Reasons To Accept:**

- Novel framework.
Introduces the first model level judge free self improvement approach for MLLMs, eliminating the need for expensive verifier models.

- Clear methodological presentation.
Provides a detailed algorithm (Algorithm 1), an explicit formula for token distribution, and illustrative figures that make the approach easy to follow.

- Strong experimental validation.
Shows consistent improvements on both the IC dataset and Object HallBenchwith far less compute.

- New benchmark contribution.
The IC dataset fills a gap by evaluating both precision and recall of captions across diverse categories, backed by human alignment studies with GPT-4o.

**Reasons To Reject:**

- Limited theoretical insight.
The paper lacks a deeper theoretical explanation for why moderate CLIP score differences yield maximal learning gains.

- Generality beyond images.
Although claimed, extension to other modalities (video, audio) is not experimentally verified.

---

> ### Author Response · Authors · 2025-06-03
>
> We thank the reviewer for the positive assessment, particularly recognizing our framework's novelty, clear presentation, strong experimental validation, and the contribution of the IC dataset. We appreciate the constructive feedback and address the points below.
>
> **1. Re. Theoretical Insight:**
> We acknowledge that our current work prioritizes the empirical demonstration and validation of our novel, efficient self-improvement framework. The observation in Figure 4, where moderate CLIP score differences between positive and negative pairs yield optimal learning gains, is indeed an interesting empirical finding. For a deep theoretical derivation, we hypothesize this relates to the quality and informativeness of preference pairs for DPO; pairs that are too similar or too disparate may offer less effective learning signals. We agree that a thorough theoretical exploration, potentially connecting our findings to concepts like "curriculum learning," "hard negative mining," or optimal "margin" in preference learning (as in recent works like Wu et al. [1] and Deng et al. [2]), is a valuable direction for future research.
>
> **2. Re. Generality Beyond Images:**
> We mentioned this point in the Section E (Limitations and Future Work) of our paper. It is generally beyond the scope of a single paper and is worth having more extensive exploration in future work. We believe that the current paper provides experimental verification for image-based MLLMs, shows encouraging results, and paves the way for the verification of other modalities in future work. Extending this to other modalities like video or audio would require modality-specific adaptations for both the `hratio` mechanism (how to blend conditional/unconditional generation) and the lightweight verifier (e.g., a video-text or audio-text encoder equivalent to CLIP).
>
> **Answering Questions:**
>
> **1. Re. CLIP Score Difference Threshold Choice and Sensitivity:**
> The CLIP score difference threshold used for potentially swapping pairs (Algorithm 1) was chosen empirically based on preliminary experiments. The goal was to find a balance: the threshold needed to be sensitive enough to catch clear instances where the initial `hratio`-based labeling was incorrect (i.e., the "positive" sample scored significantly worse than the "negative" via CLIP) but not so aggressive that it frequently overrode the intended preference signal from `hratio`. Figure 4, which analyzes DPO performance across different splits of data sorted by CLIP_score differences, provides insight into this sensitivity. It shows that performance peaks when training on pairs with moderate positive differences (e.g., Split 6). Our swapping threshold is designed to correct pairs that would fall into the detrimental large negative difference regime, effectively ensuring a cleaner dataset for DPO. A detailed ablation on the threshold value have been provided in the figure link [View Image](https://imgur.com/a/uKRriGN) (following the COLM guidance of anonymous external material).
>
> **2. Re. `hratio` Sampling Distribution:**
> We explored several strategies for `hratio` sampling, including fixed values (Appendix A, Fig. 7), a uniform distribution (Appendix A, Fig. 8), and the Gaussian distribution (Main paper, Fig. 4). The Gaussian distribution with parameters $\mu=0.5$ and $\sigma=0.15$ was selected because it yielded the best performance (F1 of 7.76 on IC dataset, vs. 7.67 for uniform and 7.70 for the best fixed pair). We hypothesize this is because:
> - Centering around $\mu=0.5$ aims to generate a substantial proportion of negative samples that introduce a moderate to significant level of hallucination, making them distinct from the low-`hratio` positive samples.
> - A $\sigma=0.15$ provides a controlled spread, ensuring a diversity of negative samples (from mildly to strongly hallucinated) without over-sampling extreme, potentially less informative cases. This diversity is generally beneficial for robust preference learning, providing a curriculum of difficulty.
>
> **3. Re. Recursive Self-Improvement:**
> This is indeed a very interesting and logical extension of our work. Applying the framework iteratively could potentially lead to further gains or reveal performance plateaus. As noted in our paper's limitations section (Section E), however, we found that the current model may get most of the benefit from the first iteration and further iterations may not get significant performance gains. It remains an open question for future exploration.
>
> We believe these more detailed clarifications and revisions will enhance the understanding and strength of our paper. We thank the reviewer again for the valuable insights.
>
> **References**
> [1] Wu, Junkang, et al. "beta-DPO: Direct Preference Optimization with Dynamic beta." Advances in Neural Information Processing Systems 37 (2024): 129944-129966.
> [2] Deng, Xun, et al. "Less is More: Improving LLM Alignment via Preference Data Selection." arXiv preprint arXiv:2502.14560 (2025).

---

### Decision · Program_Chairs · 2025-07-08

**Decision:**

Accept

**Comment:**

This paper introduces an efficient, judge-free self-improvement method for multimodal large language models (MLLMs), specifically targeting hallucination reduction through controlled negative sample generation and lightweight CLIP-based verification. The reviewers recognized its novelty, simplicity, and practical efficiency, alongside strong empirical results on relevant benchmarks. Although some concerns were raised regarding theoretical depth and hyperparameter sensitivity, the authors provided thorough clarifications and additional experiments, which largely addressed reviewers' concerns. Overall, this submission offers valuable insights and demonstrates clear benefits, making it suitable for acceptance.